# Improvement of the Collection, Maintenance, and Analysis of Neoplastic Cells from Urine Specimens with the Use of CytoMatrix

**DOI:** 10.3390/mps4030065

**Published:** 2021-09-10

**Authors:** Simone Minasi, Daniela Bosco, Bernardo Moretti, Felice Giangaspero, Antonio Santoro, Francesca Romana Buttarelli

**Affiliations:** 1Department of Radiological, Oncological and Anatomo-Pathological Sciences, Sapienza University of Rome, 00161 Roma, Italy; daniela.bosco@uniroma1.it (D.B.); bernardomoretti@hotmail.com (B.M.); felice.giangaspero@uniroma1.it (F.G.); francesca.buttarelli@uniroma1.it (F.R.B.); 2Istituto di Ricovero e Cura a Carattere Scientifico (IRCCS) Neuromed, 86077 Pozzilli, Italy; 3UCS Diagnostic S.r.l., 00067 Rome, Italy; info@ucsdiagnostic.com

**Keywords:** urothelial carcinoma, bladder cancer, urine, UroVysion, CytoMatrix

## Abstract

Urine cytology is a non-invasive test used in combination with cystoscopy for screening and follow-up of urothelial carcinoma (UC). Although cytology can be used to efficiently identify high-grade UC, it has a lower accuracy for the diagnosis of low-grade UC or patients with presence of atypical urothelial cells (AUC). For these reasons, ancillary tests have been added to urine cytology in order to improve the accuracy. However, the poor abundance of neoplastic cells in most samples and the absence of a “tissue-like” structure remains a major challenge. We used a novel synthetic support called CytoMatrix which has the property of capturing and storing cells and micro-macro aggregates within its three-dimensional structure. The urine specimens were obtained from 12 patients: 6 with suspected urothelial neoplasia (low- and high-grade) and 6 with AUC or non-neoplastic samples. The first step is the urine samples preparation, through several centrifugation passages; the second step consists in absorbing cells on the CytoMatrix, and in the subsequent formalin fixation, standard processing and paraffin embedding to prepare FFPE-CytoMatrix block. In the final step, sections are consecutively cut, stained with hematoxylin-eosin (H&E), and analyzed via UroVysion FISH and immunohistochemistry (IHC). Using our simple and reliable protocol, we can improve the quality of urine specimens, allowing a better collection, maintenance, and analysis of cells, with the advantage of using ancillary tests to support cytological diagnosis and the advantage of storing cellular material in a FFPE-CytoMatrix block.

After urine cytology analysis, mix all remaining urine samples for CytoMatrix assay workflow, divided into the first step (**A: *Urine preparation***), the second step (**B: *CytoMatrix***), and the third step (**C: *Analysis***).

## 1. Introduction

The urothelium is a transitional epithelium, lining on the inside of the bladder, ureters, and urethra, as well as the renal pelvis. Urothelial cells can become cancerous, giving rise to the urothelial carcinoma (UC), a tumor of the bladder. According to the current scientific data, UC is divided into two major groups, low-grade and high-grade, based on different pathogenetic pathways and biological behavior [1,2,3,4].

At present, the uniform and accepted classification for reporting urinary cytology, established by the Paris system and based on standardized morphological criteria, includes six different classes, reported as: (1) inadequate sample, (2) negative sample for high-grade urothelial carcinoma, (3) sample with presence of atypical urothelial cells (AUC), (4) sample with low-grade urothelial carcinoma (LGUC), (5) sample with suspicious high-grade urothelial carcinoma (SHGUC), and (6) sample with high-grade urothelial carcinoma (HGUC) [1,5] According to the European Association of Urology (EAU), the urinary cytology, in combination with cystoscopy, is a useful tool in the detection and follow-up of high grade bladder tumors but exhibits low sensitivity for low grade cancers [6,7]. As previously described, a positive cytology test can indicate the presence of UC in the urinary tract, however, a negative cytology cannot exclude its presence [7]. Moreover, it is dependent on the cytologist experience and interpretation, leading to variability in morphologic evaluation of cells [6,7,8,9].

The urine cytology samples have also several limitations due to sample variability, which could include low or inadequate cellularity, cellular degeneration before fixation and time-preservation issues. Few studies have demonstrated that urine cytology has a disappointingly low sensitivity for bladder cancer detection and the improvement of laboratory tests are needed [10,11].

For these reasons, ancillary tests have been used for urine cytology, but only a few are currently approved by both Food and Drugs Administration (FDA) and European Association of Urology guidelines to be used in laboratory, as UroVysion Bladder Cancer Kit (Abbott Molecular, Des Plaines, IL, USA) [7,12]. The UroVysion test is designed to detect aneuploidy for chromosomes 3, 7, 17, and loss of the 9p21 locus by a multicolor fluorescence in situ hybridization (FISH) technique in urine specimens from cases for recurrence monitoring in patients previously diagnosed with bladder cancer; the aneuploidy of these chromosomes is demonstrated to be useful to aid in the detection of bladder cancer [13,14,15,16,17,18]. Moreover, immunohistochemistry (IHC) can be a valuable test for some problematic samples; an IHC panel that includes biomarkers as Ki-67, cytokeratin 20 (CK20), and p53 can be quite helpful whether morphologic evaluation alone is insufficient [19]. Several studies showed the clinical utility of CK20 and Ki67 as a potential low-cost adjunct to urine cytology in diagnosis of LGUC, increasing the identification of low-grade urothelial carcinoma [20,21,22].

In cytology, one of the major difficulties is the identification of neoplastic cells from the urine samples, given their poor abundance and the lack of a tissue-architecture. Furthermore, the feasibility and the success of using ancillary tests depend on the availability of sufficient cellular material; however, it is common in daily practice that cells from urine samples are not enough to perform other tests in addition to cytology. 

In order to improve the collection, maintenance, and analysis of neoplastic cells from urine specimens, we used an innovative synthetic matrix called CytoMatrix which has the property of capturing and storing biological material, as cells and micro-macro cellular aggregates, within its three-dimensional structure from various type of samples, as fine-needle aspiration and other biologic fluids [23,24,25,26]. The use of the CytoMatrix is simple and reliable; it involves the transfer of cellular material into the CytoMatrix, the fixation in buffered formalin, the standard processing and paraffin embedding of the complex CytoMatrix-sample, and the application on the obtained inclusion of ancillary tests. 

In this study, we developed a novel CytoMatrix method and workflow for urine samples, including IHC for CK20, Ki67 and p53, followed by UroVysion FISH, in addition to the conventional hematoxylin-eosin (H&E) staining. Our preliminary results showed that this novel CytoMatrix protocol enables the rapid characterization of urothelial cancer cells after the urine specimens preparation, with the advantage of using ancillary tests to support cytological diagnosis and the advantage of storing cellular material in a formalin-fixed paraffin embedded (FFPE) block for further analysis.

## 2. Materials and Reagents

### 2.1. Materials

I.SuperFrost Plus microscope slides (Thermo Fisher Scientific, Waltham, MA, USA, Cat. no.: 10149870)II.Square Cover Slip (18 × 18 mm) (Thermo Fisher Scientific, Cat. no.: 18 × 18-1.5)III.Microscope Cover Slip Menzel (12 × 12 mm) (Thermo Fisher Scientific Menzel, Waltham, MA, USA, Cat. no.: 11708701)IV.Conical centrifuge tubes 50 mL (Thermo Fisher Scientific, Cat. no.: 10788561)V.Conical centrifuge tubes 15 mL (Thermo Fisher Scientific, Cat. no.: 10136120)VI.Eppendorf Safe-Lock Tubes 1.5 mL (Eppendorf, Hamburg, Germany, Cat. no.: 0030120086)VII.Filter tips for Gilson PIPETMAN (Gilson, city, state abbrev if USA, country, models: P20, P200, P1000)VIII.SafeCapsule—Blue screwcap container, prefilled with buffer solution (Diapath, Martinengo, Italy, Cat. no.: SC041)IX.SafeCapsule—Safety red capsule, prefilled with formalin (Diapath, Cat. no.: SC022) to freshly prepare 10% neutral buffered formalin fixativeX.Bond Polymer Refine Detection (Leica Biosystems, 1700 Leider Lane Buffalo Grove, IL, USA, Cat. no.: DS9800)XI.Primary Antibody Diluent (Leica Biosystems, Cat. no.: AR9352)XII.BOND Epitope Retrieval Solution 1 (Leica Biosystems, Cat. no.: AR9961)XIII.Dewax Solution (Leica Biosystems, Cat. no.: AR9222)XIV.Wash Solution 10×Concentrate (Leica Biosystems, Cat. no.: AR9590)XV.BOND Open Containers 30 mL (Leica Biosystems, Cat. no.: OP309700)XVI.BOND Universal Covertile (Leica Biosystems, Cat. no.: S21.2001)XVII.BOND Slide Tray (Leica Biosystems, United States, Cat. no.: S21.4586.A)XVIII.BOND™ Ready-to-Use Primary Antibody Ki67 (MM1) (Leica Biosystems, Cat. no.: PA0118)XIX.BOND™ Ready-to-Use Primary Antibody Cytokeratin 20 (Ks20.8) (Leica Biosystems, Cat. no.: PA0037)XX.BOND™ Ready-to-Use Primary Antibody p53 (DO-7) (Leica Biosystems, Cat. no.: PA0057)XXI.Vysis FISH Pretreatment Reagent Kit (Abbott Molecular, Thermo Fisher Scientific, Cat. no.: 02J03-032)XXIIUroVysion Bladder Cancer Kit (Abbott Molecular, Thermo Fisher Scientific, Cat. no.: 02J27-020)XXIII.ProbeChek UroVysion Bladder Cancer Kit Control Slides (Abbott Molecular, Thermo Fisher Scientific, Cat. no.: 02J27-011)XXIV.Rubber cement Fixogum 125 mg (Marabu, Tamm, Germany, Cat. no.: 02J27-011Marabu-125)XXV.CytoFoam Core (Bioptica, Milano, Italy, Cat. no.: CFC1)XXVI.CytoFoam Disk (Bioptica, Cat. no.: CFD1)XXVII.CytoMatrix (UCS Diagnostic, Roche Diagnostics SPA, Basel, Switzerland, Cat. no.: nd)

### 2.2. Equipment

XXVIII.Gilson™ PIPETMAN Classic™ Pipets (Gilson, models: P20, P200, P1000, Cat. no.: F123600, F123601, F123602)XXIX.Plastic Coplin Staining Jars 40 mm (Thermo Fisher Scientific, Cat. no.: 19-4)XXX.Glass Coplin Jar holds 5 slides (10 back to back) (Thermo Fisher Scientific, Cat. no.: E94)XXXI.HistoCore PELORIS 3 Premium Tissue Processing System (Leica Biosystems, model/Cat. no.: 45.0001)XXXII.Slide microtome for biological applications (Microm, Thermo Fisher Scientific, Cat. no.: model/Cat. no.: HM400R)XXXIII.Cold Plate (Kaltek, Padova, Italy, Cat. no.: CP 503)XXXIV.Histological section water bath (Weinkauf Medizintechnik, Hallerndorf, Germany, model/Cat. no.: WBRL 20)XXXV.Heated Paraffin Embedding Module (Leica Biosystems, model/Cat. no.: EG1150 H)XXXVI.Fully Automated IHC and ISH Stainer (Leica Biosystems, model/Cat. no.: BOND-III)XXXVII.Centrifuge Heraeus Primo Biofuge Primo (Thermo Fisher Scientific, Cat. no.: 75005181)XXXVIII.Water baths (37 °C and 100 °C) (Clifton, Nickel Electro Ltd, Weston-super-Mare, UK, model/Cat. no.: NE2-22D)XXXIX.Automatic Slide Hybridizer ThermoBrite (TopBrite, Locarno, Switzerland, model/Cat. no.: AS-05010-00)XL.Optical microscope (Nikon, Tokyo, Japan, model/Cat. no.: Eclipse 50i)XLI.Fluorescence microscope (Carl Zeiss, Oberkochen, Germany, model/Cat. no.: Axio Imager M1), equipped with appropriate excitation and emission filters allowing visualization of the intense red, green, aqua, and gold fluorescent signals

### 2.3. Software

XLII.AxionVision 4V, 4.8.2.0 (Carl Zeiss Micro Imaging, cOberkochen, Germany)XLIII.Imaging Software NIS-Elements F Ver4.60.00 for 64bit edition (Nikon)XLIV.ImageJ64 software (National Institutes of Health, Bethesda, MD, USA)

## 3. Procedure

### 3.1. Patients–Study Cohort

The study was conducted in the Anatomo-pathology Department of Sapienza/Umberto I (Rome, Italy), after obtaining the informed consent from all individual participants included in the study.

Fresh three timed urine specimens were obtained from 12 patients. Before the first morning sample, the subjects were instructed to drink water and liquids to induce diuresis. Each fresh sample was collected in a 50 mL centrifuge tube for three consecutive days and stored at 4 °C. 

The first urine sample was analyzed with the laboratory’s standard procedure for urine cytology, according to “Paris Classification for Urine Cytology”, and the morphological diagnosis was made by an expert cytologist (DB) [1]. Clinical data, cytological diagnosis, results of all the ancillary tests, and CytoMatrix diagnosis are recorded for each patient and shown in Table 1.

### 3.2. Urine Sample Preparation

Before starting: aPrepare Methanol-Carnoy fixative (3:1 mixture of methanol and glacial acetic acid).b.Prepare PBS Solution 1X, if not available.
After urine cytology, mix all the remaining urine specimens (stored a 4 °C) of each patient in a 50 mL centrifuge tube.Centrifuge to 600 g for 10 min at room temperature (RT) (15–30 °C).Discard the fluid, leaving about 1 mL of liquid with cellular pellet on the bottom.Resuspend the pellet in the remaining 1 mL.OPTIONAL STEP Pellets from the urine of three consecutive days from the same patient can be mixed.Add PBS 1X up to 10 mL to rinse the pellet.Centrifuge to 600 g for 10 min at RT.Discard the fluid, leaving about 0.5 mL of liquid with cellular pellet on the bottom.Add slowly 5 mL of fresh Methanol-Carnoy fixative and mix.Leave cell to fix for 1–3 h at RT.
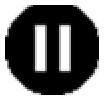
 PAUSE STEP: After this step, samples can be stored at −20 °C for the night or for alonger time.Centrifuge to 600 g for 10 min at RT.Carefully remove the supernatant and resuspend in 1–5 mL of PBS 1X to wash.OPTIONAL STEP: If the cellular pellet is very small and difficult to see, carefully remove the maximum amount of Carnoy, leaving about 50–100 μL of fixative solution to directly pipette on the matrix during CytoMatrix preparation.Centrifuge to 600 g for 10 min at RT.Carefully remove the supernatant and resuspend in 50–100 µL of PBS 1X.

### 3.3. CytoMatrix 

Pipette 50 µL of resuspended cellular pellet onto the center of the CytoMatrix (Figure 1A).


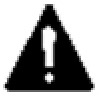
 CRITICAL STEP: The deposition of 50 µL of resuspended cellular pellet could lead to the formation of a small drop, which could reduce the absorption of the sample by the matrix. It is suggested to spread the drop over the entire Cytomatrix surface with a needle.

2.Let the cell suspension to absorb for 10–30 min, or until the matrix has completely absorbed the fluid.

OPTIONAL STEP: In case of very small amount of cells, pipet 100 uL of buffered formalin directly on the CytoMatrix surface after the complete absorption of the sample.

3.Close the biocassette (previously labelled with the patient’s id number) and immerse it in 10% buffered formalin for at least 8 h (Figure 1B).


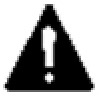
 CRITICAL STEP: It is important to recognize the side of the matrix on which cells were pipetted; the correct side is indicated by the presence of a frame.

4.Process the CytoMatrix-sample complex with the biocassette in the automatic tissue processor for FFPE, as any tissue sample (Figure 1C).


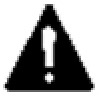
 CRITICAL STEP: Processing steps are 70% ethanol 1 h, 95% 1 h, first absolute ethanol 1 h, second absolute ethanol 1 h, third absolute ethanol 1 h, fourth absolute ethanol 2 h, first clearing agent (xylene) 1 h, second xylene 1 h.

5.Remove the CytoMatrix-sample from the processor and insert it in the paraffin-block preparation tray for paraffin infiltration (Figure 1D).


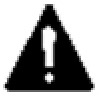
 CRITICAL STEP: During this phase, remember to orient the part facing the frame towards the bottom of the tray.

6.Add the molten paraffin at 56 °C in the tray with CytoMatrix-sample, dispensing from the Paraffin Embedding Module.7.Press gently the matrix flat to the bottom with a metallic stick, giving the correct orientation (Figure 1E).8.Transfer the tray to the cold plate and allow the paraffin to cool for 60 min.9.When the wax is completely cooled and hardened, extract the paraffin block from the tray (Figure 1F). The FFPE-CytoMatrix block is ready to cut the sections on microtome, as a standard FFPE block.10.Grind very carefully the FFPE-CytoMatrix block until it reaches the surface of the matrix.11.Pick the section up with forceps or a fine paint brush and float it on the surface of a water bath.12.Float the section on the surface of clean positively charged slides.13.Place the slide with the paraffin section on a 50 °C water bath for few seconds to bind and spread the CytoMatrix section to the glass.14.Use the first CytoMatrix slide for H&E staining.


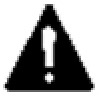
 CRITICAL STEP: For H&E staining, place slides in oven at 70 °C for 10 min, deparaffinize and rehydrate sections (3 × 3 min xylene and 3 × 3 min 100% ethanol, 1 × 3 min 95% ethanol, 1 × 3 min 70% ethanol 1 × 5 min deionized H_2_O), Hematoxylin staining 1 × 2 min, rinse with tap deionized water 1 × 5 min, Eosin staining 1 × 30 s, dehydration (1 × 2 min 70% ethanol, 1 × 2 min 90% ethanol, 3 × 2 min 100% ethanol), 3 × 5 min xylene, Coverslip slides using Permount (xylene based).

15.Observe the H&E on optical microscope in order to evaluate the morphology and the amount of cells for each patient.16.Only in cases with sufficient material, cut the second 5 µm section for UroVysion FISH.17.Cut the third, fourth and fifth 3 µm sections for immunohistochemistry.

The entire protocol from urine sample to FFPE-CytoMatrix block can be viewed in Appendix A.

### 3.4. UroVysion FISH

As reagents, we used:a.The UroVysion Bladder Cancer Kit (Abbott Molecular) which contains DNA Probe Mixture (Fluorophore-labelled DNA probes for chromosomes 3, 7, and 17, and locus 9p21 in hybridization buffer), DAPI II Counterstain, 20X SSC, NP-40 (non-ionic detergent).b.The Vysis FISH Pretreatment Reagent Kit (Abbott Molecular) which includes Vysis Protease solution (3 × 25 mg) with Pepsin Activity, Vysis Pepsin Buffer (3 × 50 mL) 10 mM HCl, Vysis PBS 1X (2 × 250 mL), Vysis 20X SSC (66 g), 10% neutral buffered formalin and 4’,6-Diamidino-2-phenylindole (DAPI) solution.Before starting:c.In each session, include a Probe Check Bladder Cells Control Slides, containing both a positive control and a negative control on the same slide.
a.Prepare Ethanol solutions at 70%, 90%, and 100% with purified waters in coplin jars.b.Prepare solutions 20X SCC (pH: 5.3), 2X SCC, 0.4X SSC/0.3% NP-40, 2X SSC/0.1% NP-40.c.Add protease solution to Coplin jar and place in a 39 °C water bath for at least 30 min before use, or until the solution temperature reaches 37 ± 1 °C. Verify the solution temperature before use with a calibrated thermometer.d.Add 2X SCC solution to Coplin jar and place in a 94 °C water bath for at least 30 min before use, or until the solution temperature reaches 94 ± 1 °C. Verify the solution temperature before use with a calibrated thermometer.
Select the hybridization area with a circle in the 5 µm sections cut from FFPE-CytoMatrix blocks for FISH analysis.Place slides in oven at 70 °C for 30 min.Deparaffinize in xylene for 3 × 10 min.Rehydrate sections in 100% ethanol 2 × 10 min, 90% ethanol 1 × 5 min, 70% ethanol 1 × 3 min, and deionized H_2_O 1 × 3 min.Allow slides to completely dry at RT.Immerse slides in 2X SSC for 10 min at 94 ± 1 °C.Wash with deionized H_2_O for 3 min.Immerse slides in protease solution for 30 min at 37 ± 1 °C.Wash slides in 1X PBS for 3 min.Dehydrate sections in 70% ethanol 2 min, 90% ethanol 2 min, 100% ethanol 2 min.Allow slides to completely dry at RT.Probes preparation:a.Remove the UroVysion probe from −20 °C storage and allow to warm to RT.b.Vortex to mix and then spin the tubes briefly (3 s) in a microcentrifuge to bring the contents to the bottom of the tube.c.Apply 3–5 μL of probes solution to the selected target area of each slide.d.Immediately, place a 12–18 mm round glass coverslip over the probe.e.Carefully apply light pressure on the coverslip to allow the probe solution to spread evenly under the coverslip (Note: air bubbles can interfere with hybridization and should be avoided).f.Seal coverslip with rubber cement around the periphery of the coverslip.Add around 75 mL of distilled or deionized water in each humid chamber of the Automatic Slide Hybridizer ThermoBrite.Set the program for denaturation temperature 76 °C 5 min (Denaturation Step) and for hybridization temperature 39 °C 14–18 h (Hybridization Step) and run it.Post-Hybridization washes:a.Fill a Coplin jar with 0.4X SSC/0.3% NP-40 solution and place it in a 74 °C water bath for at least 30 min before use, or until the solution temperature reaches 73 ± 1 °C. Verify the solution temperature before use with a calibrated thermometer.b.Fill a second Coplin jar with 2X SSC/0.1% NP-40 solution and place it at RT.c.Remove the rubber cement and the coverslip from the slides with a metallic tweezer (Note: if the coverslip does not come off easily, immerse the slides in 2X SSC/0.1% NP-40 solution for a few seconds, in order to easily remove it).d.Immediately after removing the coverslip, place the slides in the Coplin jar with 0.4X SSC/0.3% NP-40 solution at 73 ± 1 °C and incubate for 2 min.e.Remove the slides from the wash solution, place the slides in the Coplin jar containing 2X SSC/0.1% NP-40 at RT and incubate for 1 min.f.Remove the slides from the second wash solution and place vertically in a dark area on a paper towel to dry completely.Apply 10 μL of DAPI II onto the target area and place a coverslip (18 mm square is recommended), avoiding air bubbles.Store the slides in the dark prior to signal enumeration under the fluorescence microscope.




### 3.5. Immunohistochemistry

The evaluation of protein expression was carried out by streptavidin–biotin–immunoperoxidase technique on 3 μm sections from FFPE-CytoMatrix blocks. Immunohistochemical analysis was performed on an automated immunohistochemical stainer Leica Bond III (Dept. of Radiological, Oncological and Anatomo-Pathological Sciences, Sapienza University of Rome), according to manufacturer’s instructions.

We used the BOND reagents and the three primary antibodies (Anti-Ki67, Anti-CK20, and Anti-p53) specified in *Materials* Section.

Place the three labelled slides with 3 µm sections in oven at 70 °C for 15 min.Set the program on the automatic stainer according to manufacturer’s instructions. The protocol used for each antibody is shown in Appendix A.Load the slides onto the BOND Slide Tray, apply new BOND Universal Covertiles, place it in the Leica BOND III and start the run.When the run is complete, remove Covertiles, rinse the slides in deionized water, dehydrate in ethanol, clear in xylene and mount sections with coverslips.Store the slides in a slide-box prior to evaluate under the optical microscope.

## 4. Results

The CytoMatrix is a matrix based on chitosan, a natural non-synthetic polymer. The CytoMatrix porosity and ionic strength makes it perfect for attracting and maintaining cells [23,24,25,26]. After the loading of cells from urine, the fixation, processing, and paraffin embedding of the complex CytoMatrix-sample, we cut on microtome the first 3 µm section from FFPE-CytoMatrix block and we used it for H&E staining. H&E are used to assess the number and the morphology of cells on optical microscope. 

Results can briefly summarized as follows: 12/14 (85%) tested urine specimens, obtained from different patients, showed sufficient cellular material after the CytoMatrix protocol and H&E staining (Figure 2A). The two cases with insufficient materials were originally diagnosed as “inadequate sample”, given that not enough cells were found in the first urine sample analyzed by cytology, suggesting that cells were absent or too few to be captured and stored by CytoMatrix as well. For all other samples, even starting from a limited number of cells, it was possible to obtain from each single CytoMatrix several slides to perform the FISH analysis for UroVysion and the panel of immunostainings for various markers.Our tests showed that the first ten 3 μm sections are the richest in cellular material; on average, the cells penetrate inside the matrix for about 30 μm. In subsequent cuts, the cells are poor and lose a “tissue-like” structure.The fibers of the CytoMatrix are stained fuchsia by eosin and can be clearly distinguished.H&E staining showed the power of this tool to confer a 3D-structure of cells from urine. CytoMatrix-FFPE block from patient number 4, stained with H&E, clearly showed the presence of papillary clusters, small vessels and a fibrous-connective axis (Figure 2B); in this case, the property of CytoMatrix allowed to capture and store cells and also micro-cellular aggregates in its three-dimensional structure, helping the diagnosis of HGUC.CytoFoam is a 12-mm disc previously described in literature and used to adsorb various cytological samples, as fine needle aspiration of thyroid carcinomas or plasma for the evaluation of circulating tumor cells [27,28]. It is a diagnostic tool similar to CytoMatrix, since it is processed as a tissue fragment, formalin-fixed, paraffin-embedded, and used for IHC or other molecular analyses. We used CytoFoam, as previously described, to absorb our cell suspensions from urine specimens in order to compare the results obtained with CytoMatrix [27,28]. Interestingly, when compared with this CytoFoam adsorbent support, CytoMatrix can improve the amount of cells and can also confer an architecture or a "tissue-like" structure (Figure 2C).

### 4.1. CytoMatrix-FFPE UroVysion Evaluation

Details regarding the cells selection criteria and the probes evaluation criteria have been provided by the manufacturer (Abbott Molecular) and by several studies [16] [17]. Briefly:UroVysion probe signals and DAPI counterstain should be viewed under fluorescence microscope with the following filters: DAPI single-bandpass, Aqua single-bandpass (chromosome 17), Yellow (Gold) single-bandpass (9p21 locus), Red/Green dual-bandpass (chromosomes 3 and 7).Begin analysis in the upper left quadrant of the target area. Scan fields from left to right and top to bottom, without rescanning the same areas.Use the following criteria to select cells suspicious for malignancy: large nuclear size, irregular nuclear shape, “patchy” DAPI staining, and cell clusters.Determine the number of signals for all 4 probes in a minimum of 200 cells and/or 25 morphologically abnormal neoplastic cells.Record the chromosome pattern only if:
There is a gain (3 or more signals) of 2 or more of chromosomes 3 (red), 7 (green), or 17 (aqua).There is a loss of both copies of LSI 9p21 (Gold).Other situations should be considered uninterpretable due to hybridization failure.Take images using a Zeiss fluorescence microscope (Axio Imager M1), equipped with appropriate excitation and emission filters, at magnification 20x and 100x within the area of interest.Analyze the images with Zeiss AxionVision 4V, 4.8.2.0 Software, manually counting the number of cells in which signals colocalize with DAPI nuclei.If no abnormalities are detected, the remaining cells are counted until a sufficient number of cells without chromosomal abnormalities are visualized (minimum 200 cells evaluated).A positive result is indicated by the presence of ≥4 cells with gains of two or more of chromosomes 3, 7 and 17 on 25 neoplastic cells analyzed. In the case of chromosome 9p21, a positive result is considered when ≥12 cells show absence of 9p21 signals on 25 neoplastic cells analyzed.Our results showed that:The UroVysion test can be used with CytoMatrix samples and can improve the accuracy of diagnosis for urothelial carcinoma.All non-neoplastic samples (5/5, 100%) showed normal copy number of chromosomes 3, 7, 17 and 9p21 (Figure 3A).All LGUC and HGUC (7/7, 100%) showed the presence of nuclei with 3 or more signals for chr 3, 7, and 17 in selected areas; only 2 HGUC showed homozygous deletion of 9p21, not present in LGUC, suggesting that p16/INK4 deletion could be associated higher malignancy and aggressiveness of urothelial tumors (Figure 3A).The fibers of the CytoMatrix exhibited an autofluorescence that can be clearly distinguished under the microscope with all the filters (Figure 3B).


### 4.2. CytoMatrix-FFPE IHC Evaluation

After the automatic IHC for the antibodies Anti-Ki67, Anti-CK20, and Anti-p53, slides from FFPE-CytoMatrix block are visualized under optical microscope to evaluate the expression of these proteins. The evaluation criteria are:The slides are scored on the scale 0–3+ on the basis of DAB intensity and are quantified by counting at least 200 stained cells.Take images using a Nikon optical microscope (Eclipse 50i) at magnification 10x, 20x and 40x within the area of interest (Note: all microscope and camera settings (e.g., light level, exposure, gain, etc.) should be identical for all images).Analyze the images with Nikon Imaging Software NIS-Elements and quantify the DAB signal with ImageJ64 software.Manually count the number of positive cells with the expression of the antigen of interest.Staining intensity is scored as “no expression” (0), “weak expression” (1+), “moderate expression” (2+) and “strong expression” (3+).Ki67 is scored from 0% to 30% of nuclei, to evaluate proliferation rate.CK20 is scored as positive when neoplastic cells show cytoplasmatic/membrane staining intensity >1, showing the presence of cells with epithelial origin.p53 is scored as positive when neoplastic cells show nuclear staining intensity >1, suggesting the presence of a TP53 mutation.

Our results showed that:
The Automatic IHC can be used with CytoMatrix samples and can improve the accuracy of diagnosis for urothelial carcinoma.The fibers of the CytoMatrix did not show any staining in IHC under the microscope.Ki67 staining was positive in neoplastic cells of two HGUC (N° 2, 4), while the other HGUC (N° 1, 3) showed negative staining. Staining of Ki67 was negative in LGUC and non-neoplastic cases; however, two non-neoplastic samples (N° 10, 11) showed weak positivity for Ki67 (<5% of cells) in granulocytes and immune cells (Figure 4).CK20 staining was positive in HGUC (N° 1, 2, 3) that maintain epithelial differentiation and it can be useful to identify neoplastic cells; however, immuno-staining for CK20 can also be negative in HGUC (N° 4) with loss of epithelial differentiation. Staining of CK20 was negative in all LGUC and non-neoplastic cases (Figure 4).p53 staining was negative in all analyzed cases, both in neoplastic and non-neoplastic samples, suggesting a low utility of this biomarker in supporting the diagnosis of urothelial carcinoma.

In conclusion, our CytoMatrix assay, supported by IHC for CK20/Ki67 and UroVysion, confirmed the first cytological diagnosis for 4/4 (100%) HGUCs, 2/2 (100%) LGUCs, and 5/5 (100%) non-neoplastic samples (Table 1) (Appendix A). Interestingly, it was possible to diagnose as LGUC one sample (n° 9), previously not evaluable by cytology (low cellularity and cellular degeneration), with the use of our CytoMatrix workflow, suggesting the diagnostic potential of this support for urothelial tumors; this sample was also tested with UroVysion and showed the gain of chr 3 and 17.

Our results show that CytoMatrix allows an easy and fast characterization of urothelial cells, minimizing manual handling, with the advantage of keeping urothelial cells from urine samples in a FFPE-CytoMatrix block and also allowing the analysis of several molecular markers. 

The use of CytoMatrix allows to study chromosomal alterations with UroVysion, improving diagnostic accuracy of urothelial carcinoma. Moreover, different markers or additional tests could potentially be applicable with CytoMatrix samples.

We suggest that CytoMatrix could be a reliable tool to overcome the current limits of traditional collection and analysis methods of urine cytology, thereby improving the reliability for a conclusive diagnostic interpretation.

Although further validation on a larger number of cases will be necessary, our CytoMatrix protocol is highly applicable and might contribute in the future to improve the collection, maintenance, and analysis of cells from urine specimens. 

## Figures and Tables

**Figure 1 mps-04-00065-f001:**
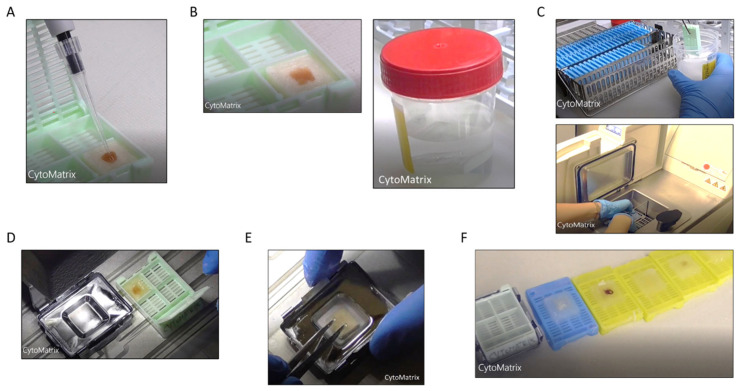
CytoMatrix preparation for urine specimens (**A**) Load the cellular suspension into the center of CytoMatrix. (**B**) Absorb and fix in buffered formalin. (**C**) Process the CytoMatrix-sample in automatic tissue processor. (**D**) Add Molten paraffin in tray. (**E**) Press the CytoMatrix flat to the bottom. (**F**) Cool and extract the FFPE-CytoMatrix blocks.

**Figure 2 mps-04-00065-f002:**
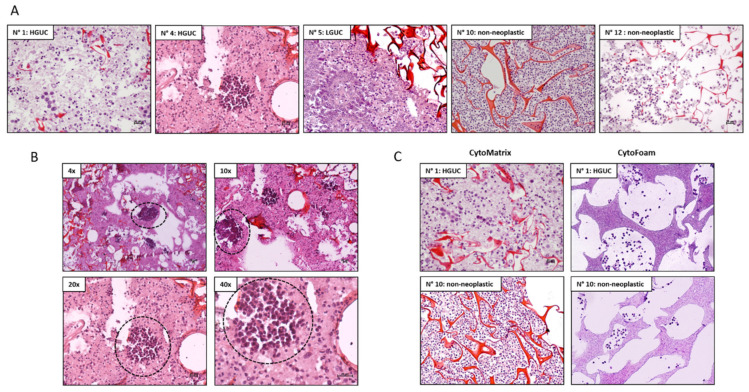
H&E staining of CytoMatrix-FFPE blocks. (**A**) Panel of H&E at magnification 20x from different representative cases diagnosed as HGUC, LGUC or non-neoplastic. (**B**) H&E of CytoMatrix-FFPE N° 4 shows the presence of a papillary cluster (circle in black) with neoplastic cells characterized by large and hyperchromatic nucleus, increased nucleus/cytoplasm ratio, and nuclear pleomorphism (at magnification 4×, 10×, 20× and 40×). (**C**) Comparison of H&E of samples N° 1 and N° 10 obtained from CytoMatrix-FFPE or CytoFoam (at magnification 20×); CytoMatrix clearly shows an increase in the number of cells and the presence of a “tissue-like” structure.

**Figure 3 mps-04-00065-f003:**
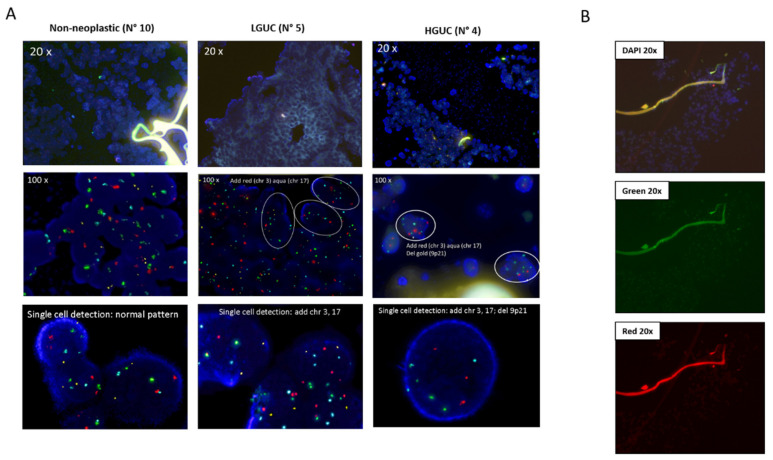
UroVysion FISH on CytoMatrix-FFPE blocks. (**A**) Panel of FISH at magnification 20x, 100x and single cell detection from different representative cases diagnosed as non-neoplastic (N°10), LGUC (N° 5), and HGUC (N° 4); images evidence the presence of 3 or more signals for chr 3 (red) and chr 17 (aqua) in >10% cells in tumor samples, and the HGUC shows also homozygous deletion of 9p21 (gold). (**B**) Autofluorescence of the CytoMatrix fibers with DAPI, Green, and Red filters (at magnification 20x).

**Figure 4 mps-04-00065-f004:**
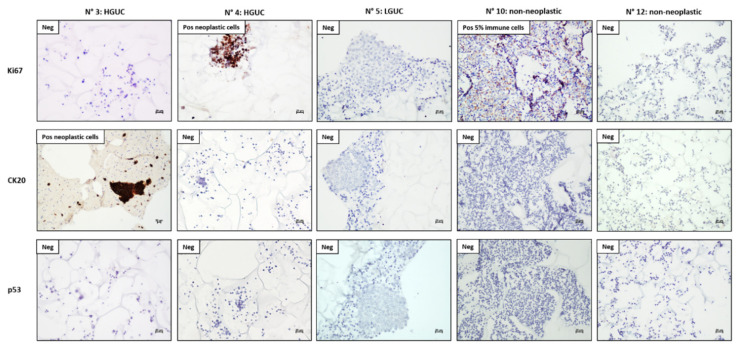
Automatic IHC on CytoMatrix-FFPE blocks. Panel of IHC at magnification 20x from different representative cases diagnosed as HGUC (N° 3 and 4), LGUC (N° 5), and non-neoplastic (N° 10 and 12); images evidence the presence of one positive cases for CK20 (N° 3), two for Ki67 (N° 4 and 10) and none for p53.

**Table 1 mps-04-00065-t001:** Clinical and molecular features of urine samples analyzed with CytoMatrix assay, including information of initial cytological diagnosis. HGUC: high-grade urothelial carcinoma; LGUC: low-grade urothelial carcinoma; AUC: sample with presence of atypical urothelial cells.

n°	Sex	Age at Diagnosis	Cytological Diagnosis	CK20	Ki67	p53	UroVysion FISH	CytoMatrix Diagnosis
1	M	90 y	HGUC	Pos > 10%	neg	neg	gain of chr 3 (red), 7 (green)	HGUC confirmed
2	M	75 y	HGUC	Pos > 10%	Pos 5%	neg	gain of chr 3, 17 (aqua), homozygous loss 9p21 (gold)	HGUC confirmed
3	M	64 y	HGUC	Pos > 10%	neg	neg	gain of chr 3, 7	HGUC confirmed
4	M	62 y	HGUC	neg	Pos 10%	neg	gain of chr 3, 7, 17, homozygous loss 9p21	HGUC confirmed
5	F	88 y	LGUC	neg	neg	neg	gain of chr 3, 17	LGUC confirmed
6	M	68 y	LGUC	neg	neg	neg	gain of chr 3, 17	LGUC confirmed
7	F	70 y	AUC	neg	neg	neg	normal copy number chr 3, 7, 17, 9p21	Non neoplastic confirmed
8	M	67 y	AUC	neg	neg	neg	normal copy number chr 3, 7, 17, 9p21	Non neoplastic confirmed
9	M	66 y	Non evaluable	Pos 10%	neg	neg	gain of chr 3, 17	LGUC newly diagnosed
10	M	83 y	Non neoplastic	neg	Pos immune cells	neg	normal copy number chr 3, 7, 17, 9p21	Non neoplastic confirmed
11	F	66 y	Non neoplastic	neg	Pos immune cells	neg	normal copy number chr 3, 7, 17, 9p21	Non neoplastic confirmed
12	M	74 y	Non neoplastic	neg	neg	neg	normal copy number chr 3, 7, 17, 9p21	Non neoplastic confirmed

## Data Availability

The data presented in this study are available on request from the corresponding author.

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
