# Peer review of "Improvement of the Collection, Maintenance, and Analysis of Neoplastic Cells from Urine Specimens with the Use of CytoMatrix"

_mps, 2021, doi:10.3390/mps4030065_

Round 1

Reviewer 1 Report

The application of this methods protocol paper is highly applicable to improve UC patient outcomes. 

It would be helpful to further elaborate on the claim of easy and fast characterization of UC (line 533-534) in comparison the current urine cytology standards. The graphical abstract is helpful, but not clear which parts of the workflow may overlap with urine cytology.

Of the 14 urine specimens tested , were the other 2 samples that did not show sufficient cellular material collected in any different manner?

In table 1, sample 11, is there a word missing in the last column?

Is CitoMatrix the same as CytoMatrix? If so, it's confusing to see two different spellings.

Can Cytoform be described in more detail?

Reviewer 2 Report

The manuscript entitled “Improvement of the collection, maintenance, and analysis of neoplastic cells from urine specimens with the use of CytoMatrix” by Minasi et al present a protocol where CytoMatrix is used to improve the collection, processing and analysis of cells from urine specimens for diagnosis of bladder cancer. The cytomatrix approach adds to the standard cytology by allowing the preparation of a FFPE block and additional tests such as FISH and IHC.

The protocol is described with enough detail to allow its replication. Critical steps and additional notes are provided to optimize its implementation. Overall, it is a useful and valuable guide. I did not have access to the video, but the table, figures and supplementary information provided are adequate.

Below I present minor points that, in my opinion, would improve the overall quality of the manuscript.

Points of improvement:

The citations in the text lack a comma between multiple reference.

Introduction

  1. In the Introduction, line 54 presents an overstatement and incomplete picture of the role of cytology for the diagnosis of bladder cancer. According to the European Association guidelines “Cytology is useful, particularly as an adjunct to cystoscopy, in patients with HG/G3 tumours. Positive voided urinary cytology can indicate an urothelial carcinoma anywhere in the urinary tract; negative cytology, however, does not exclude its presence.” Instead of presenting urine cytology as “the current standard for the diagnosis” I would suggest referring it as an auxiliary text in specific contexts. See https://uroweb.org/guideline/non-muscle-invasive-bladder-cancer/#5
  2. Since the study is conducted in Europe I would also suggest to provide context for the use of auxiliary tests recommended/approved in Europe in addition to FDA.

Materials and Reagents

General comment: extensive list including items that are of general purpose and of no added value to the current protocol such as Excel, GIMP. Consider simplifying.

Procedure

Some information seems adequate for a detailed protocol to keep in the lab but maybe of little use in a lab with different equipment. To make it a more concise and straightforward protocol, I would suggest keeping detailed information regarding the CytoMatrix setup and the main steps of the downstream analysis but reduce some of the info related with the way to run specific instruments. This comment should be view in the light of the journal guidelines for protocols.

  1. Line 234: citomatrix or cytomatrix? These 2 terms are used in different parts of the manuscript. It is unclear if they represent the same thing. Consider clarifying.
  2. In 3.3, line 235: delete “directly”, line 255: for clarity instead of saying “to orient the part with the frame” write: “orient the part facing the frame towards..”; step 15, line 277: unnecessary comma
  3. For IHC, considering that it is a standard technique and most of the steps are highly detailed and specific for a Leica Bond strainer I would suggest simplifying the protocol, i.e. the protocol for staining is relevant but all details about the way the machine is run seems unnecessary.

Results

  1. Line 422, remove comma.
  2. Line 494: proteins
  3. Line 538 in conclusion is unclear, please reformulate: “However, the number of analyzed cases need to be implemented in order to validate our preliminary results”
  4. The following statement is mentioned in several places in the manuscript ”…test can be used with CytoMatrix samples and can improve the accuracy of diagnosis for urothelial carcinoma”. Could you please discuss to what extent this is observed in your study and relate with the table 1. I agree with the authors that the CytoMatrix increases the options to perform additional tests but, with the data provided, it is not discussed to what extent it provided additional value to the diagnosis. Maybe the markers are not the most informative or there are specific situations in which additional tests would be informative.

Materials and Reagents

General comment: extensive list including items that are of general purpose and of no added value to the current protocol such as Excel, GIMP. Consider simplifying.

Procedure

Some information seems adequate for a detailed protocol to keep in the lab but maybe of little use in a lab with different equipment. To make it a more concise and straightforward protocol, I would suggest keeping detailed information regarding the CytoMatrix setup and the main steps of the downstream analysis but reduce some of the info related with the way to run specific instruments. This comment should be view in the light of the journal guidelines for protocols.

  1. Line 234: citomatrix or cytomatrix? These 2 terms are used in different parts of the manuscript. It is unclear if they represent the same thing. Consider clarifying.
  2. In 3.3, line 235: delete “directly”, line 255: for clarity instead of saying “to orient the part with the frame” write: “orient the part facing the frame towards..”; step 15, line 277: unnecessary comma
  3. For IHC, considering that it is a standard technique and most of the steps are highly detailed and specific for a Leica Bond strainer I would suggest simplifying the protocol, i.e. the protocol for staining is relevant but all details about the way the machine is run seems unnecessary.

Results

  1. Line 422, remove comma.
  2. Line 494: proteins
  3. Line 538 in conclusion is unclear, please reformulate: “However, the number of analyzed cases need to be implemented in order to validate our preliminary results”
  4. The following statement is mentioned in several places in the manuscript ”…test can be used with CytoMatrix samples and can improve the accuracy of diagnosis for urothelial carcinoma”. Could you please discuss to what extent this is observed in your study and relate with the table 1. I agree with the authors that the CytoMatrix increases the options to perform additional tests but, with the data provided, it is not discussed to what extent it provided additional value to the diagnosis. Maybe the markers are not the most informative or there are specific situations in which additional tests would be informative.
